# Leptomeningeal Metastases in Melanoma Patients: An Update on and Future Perspectives for Diagnosis and Treatment

**DOI:** 10.3390/ijms241411443

**Published:** 2023-07-14

**Authors:** Julian Steininger, Frank Friedrich Gellrich, Kay Engellandt, Matthias Meinhardt, Dana Westphal, Stefan Beissert, Friedegund Meier, Isabella C. Glitza Oliva

**Affiliations:** 1Department of Dermatology, Faculty of Medicine and University Hospital Carl Gustav Carus, Technische Universität (TU) Dresden, 01307 Dresden, Germany; frankfriedrich.gellrich@ukdd.de (F.F.G.); dana.westphal@ukdd.de (D.W.); stefan.beissert@ukdd.de (S.B.); friedegund.meier@ukdd.de (F.M.); 2Department of Neuroradiology, Faculty of Medicine and University Hospital Carl Gustav Carus, Technische Universität (TU) Dresden, 01307 Dresden, Germany; kay.engellandt@ukdd.de; 3Institute of Pathology, University Hospital Carl Gustav Carus, Technische Universität (TU) Dresden, 01307 Dresden, Germany; matthias.meinhardt@ukdd.de; 4Skin Cancer Center at the University Cancer Center, National Center for Tumor Diseases (NCT/UCC), 01307 Dresden, Germany; 5Department of Melanoma Medical Oncology, The University of Texas MD Anderson Cancer Center, Houston, TX 77030, USA; icglitza@mdanderson.org

**Keywords:** melanoma, leptomeningeal disease, CNS microenvironment, intrathecal therapy, leptomeningeal carcinomatosis, leptomeningeal metastases

## Abstract

Leptomeningeal disease (LMD) is a devastating complication of cancer with a particularly poor prognosis. Among solid tumours, malignant melanoma (MM) has one of the highest rates of metastasis to the leptomeninges, with approximately 10–15% of patients with advanced disease developing LMD. Tumour cells that metastasise to the brain have unique properties that allow them to cross the blood–brain barrier, evade the immune system, and survive in the brain microenvironment. Metastatic colonisation is achieved through dynamic communication between metastatic cells and the tumour microenvironment, resulting in a tumour-permissive milieu. Despite advances in treatment options, the incidence of LMD appears to be increasing and current treatment modalities have a limited impact on survival. This review provides an overview of the biology of LMD, diagnosis and current treatment approaches for MM patients with LMD, and an overview of ongoing clinical trials. Despite the still limited efficacy of current therapies, there is hope that emerging treatments will improve the outcomes for patients with LMD.

## 1. Introduction

Leptomeningeal disease (LMD) represents malignant seeding to the connective tissue layers of the soft meninges (arachnoid and pia mater). The most common primary tumours leading to LMD are lung cancer, breast cancer, malignant melanoma (MM), lymphoma, and leukaemia. Among solid tumours, MM has not only the highest overall risk of parenchymal central nervous system (CNS) invasion, but also one of the highest rates of metastasis to the leptomeninges, with up to 10–15% of patients with advanced disease [1]. At the time of initial diagnosis, almost 80% of patients also have both CNS and extra-CNS metastases [2]. Although rare overall, the incidence of LMD is increasing, which may be due to improved and advanced diagnostic capabilities and current therapeutic options for the metastatic disease [3]. Despite the ongoing advances and assessment of new therapies as part of clinical trials, LMD remains an unmet medical need with limited treatment, very few clinical trial options, and a correspondingly poor prognosis that has improved only marginally in recent years. In the largest to date reported case series of 178 MM patients with LMD, the median overall survival (OS) with existing contemporary therapies was only 3.5 months [2], which was also observed by other groups [4].

This review provides an overview of the biology of LMD, as well as diagnosis and current treatment approaches for MM patients with LMD. Finally, current ongoing clinical trials are reviewed. In this context, intrathecal (IT) approaches of immune checkpoint inhibitor (ICI) therapy in particular are discussed, as first demonstrated by Glitza et al. [5].

## 2. Materials and Methods

This review summarises clinically relevant data from prospective and retrospective studies on the treatment of patients with MM and LMD. We performed a systematic review of the literature using PubMed from January 2010 to February 2023 for all published articles on “leptomeningeal disease”, “leptomeningeal metastases”, and “leptomeningeal carcinomatosis” in combination with “melanoma”. We included English-language articles with a focus on previous reviews and clinical trials.

## 3. Anatomical Structure

The leptomeninges are thin membranes that cover the brain and spinal cord. They consist of the pia mater and the arachnoid mater, with the subarachnoid space located between them, which contains the cerebrospinal fluid (CSF). The pia mater is the innermost layer of the leptomeninges, adhering directly to the surface of the brain and spinal cord. It is highly vascularised and contains many small blood vessels that supply nutrients and oxygen to the brain. The arachnoid mater is the middle layer of the meninges and is separated from the pia mater by the subarachnoid space. It is a thin, avascular membrane that surrounds the subarachnoid space, which is filled with CSF. The subarachnoid space, between the arachnoid mater and the pia mater, extends throughout the brain and spinal cord and provides a cushioning effect to protect from trauma.

Cancer cells have the ability to enter CSF via four main routes (Figure 1): First, they can enter via the arterial circulation by passing through the choroid plexus [6]. Second, they can travel through Bateson’s plexus or bridging veins and enter via the venous circulation [7]. Third, cancer cells may follow the cranial nerves or spinal roots perineurally to infiltrate CSF [6]. Finally, they can penetrate the glia limitans from the brain parenchyma and gain access to CSF [8]. In addition, another possibility is the iatrogenic transfer of tumour cells during surgical resection of brain metastases [9]. In MM, LMD is largely attributed to metastatic dissemination as well, although it can very rarely occur by malignant transformation of pre-existing melanocytes within the leptomeningeal space [10,11,12].

Clinical observations and autopsy studies have shown that leptomeningeal cancer cells occur in two opposite phenotypes—floating freely in CSF or adhering to the leptomeninges [13]. Adherent cells attach to surfaces, grow flat, and spread out, while floating cells do not adhere and grow spherically instead. Remsik et al. demonstrated that the floating type represents the more aggressive form of LMD [14]. In vitro, tumour cells of this type contain less ATP and slow their growth in both adherent and non-adherent settings. When implanted in vivo, they colonise the subarachnoid space of mice more rapidly, accelerating neurological symptoms and death. The authors concluded that CSF with floating tumour cells resembles late-stage tumours, both of which have low nutrient levels and hypoxia. Preserved transcriptomic adaptations of free-floating cells may facilitate survival in vivo, but are a disadvantage in regular in vitro culture. These findings are also supported by real-world data, as the positive detection of tumour cells in CSF, which mainly represents the free-floating tumour type, correlates with a worse prognosis [2].

**Figure 1 ijms-24-11443-f001:**
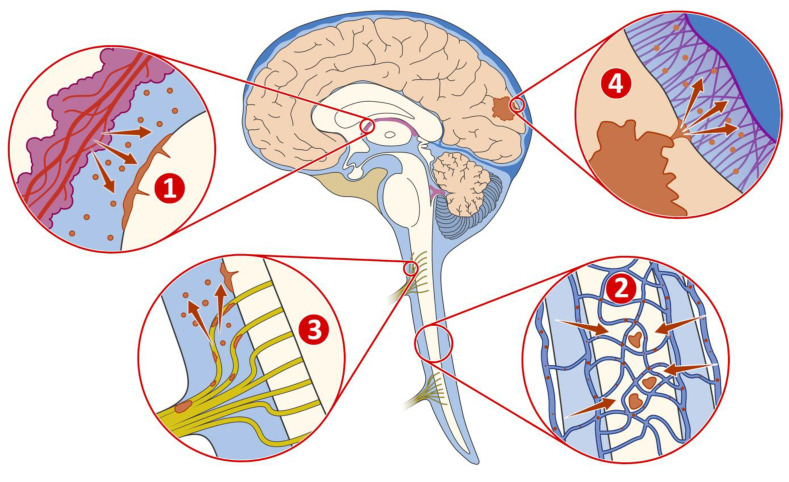
Four main routes of cancer cells to enter CSF. 1: Via the arterial circulation by passing through the choroid plexus. 2: Via the venous circulation through Bateson’s plexus or bridging veins. 3: Via the cranial nerves or spinal roots perineurally. 4: Via penetrating the glia limitans from the brain parenchyma. Figure adapted from Saadeh et al. [15] and created by F.F. Gellrich.

## 4. Tumour Microenvironment

A growing body of evidence suggests that the tumour microenvironment (TME) is actively involved in pathogenesis and treatment response [16], leading to the newly accepted concept that tumours are holistic, complex biosystems rather than a collection of malignant cancer cells [17]. Innate and adaptive immune cells, blood and lymphatic vessel networks, and stromal cells are common components of the TME [18]. Recently, tumours have also been reported to harbour bacteria, raising the possibility that the microbiome may be another player in the complex tumour ecosystem. For example, recent evidence suggests that the gut microbiome may have an impact on both the response and toxicity of cancer therapies such as immunotherapy [19].

At the core of the TME, tumour cells control the function of cellular and non-cellular components through complex signalling networks to exploit non-malignant cells for their own benefit [20]. The components of the TME are emerging as critical mediators of metastatic colonisation and tumour progression. Although organ-specific patterns of progression and therapeutic response are poorly understood, it is clear that each metastatic site has its own unique immunological microenvironment [21]. Tumour cells that metastasise to the brain possess properties that enable them to cross the blood–brain barrier (BBB), evade the immune system, and survive in the unique microenvironment of the brain [22]. Metastatic colonisation is achieved through complex and dynamic communication between the metastatic cells and the surrounding TME, resulting in a tumour-permissive milieu.

Compared to other major metastatic sites, the CSF TME is still poorly understood but appears to be very different, being almost acellular and low in protein, glucose, and cytokines [23]. Because of its barren composition, it is thought that invading tumour cells are able to optimise the CSF landscape in their favour.

Boire et al. showed that cancer cells in CSF upregulate complement component 3 (C3) to activate the C3a receptor in the choroid plexus epithelium, thereby disrupting the blood–CSF barrier [24]. This allows various plasma components, such as the EGFR ligand amphiregulin, to enter CSF and sustain cancer cell growth. Treatment response is associated with a decrease in CSF amphiregulin levels in patients with elevated CSF amphiregulin levels. As C3 upregulation in primary tumours has been shown to be predictive for LMD, its medical inhibition may prevent metastatic seeding to this space. In a recently published paper, Smalley et al. showed that the TME of LMD is characterised by an immunosuppressive T cell landscape and thus differs from solid cerebral and cutaneous metastases [25]. The cellular composition of cerebral metastases was found to be similar to that of cutaneous metastases, whereas that of LMD differed by a high proportion of dysfunctional CD4+ and CD8+ cells and a low proportion of B cells. In an LMD patient with long-term survival of 38 months, the immunological environment of LMD was shown to be more similar to the CSF of control patients without LMD. In addition to CSF in LMD containing high levels of complement (e.g., C3, as also described by Boire et al.), other components of the innate immune system and immunosuppressive growth factors such as TGF beta have been identified [26]. One hypothesis is that circulating factors lead to reprogramming of T cells in the leptomeningeal milieu, which then enter a dysfunctional state. It has also been shown that a novel population of dendritic cells (DC3) correlates with increased OS, independent of disease site and treatment [25]. The presence of these DC3s positively regulates the immune environment in both patient samples and preclinical melanoma models by modulating activated T cells and MHC expression in the tumour.

In order to gain a better perspective of the LMD TME, researchers from H. Lee Moffitt Cancer Center and Research Institute recently established a preclinical model by growing patient-derived CSF circulating tumour cells (CTCs) in vitro and in vivo [27]: CTCs from the CSF of LMD patients were detected by an adapted version of the CellSearch^®^ platform, using the CELLTRACKS Circulating Melanoma Cell Kit and cultured ex vivo in human meningeal cell-conditioned medium (containing secreted growth factors, GM-CSF, and VEGF-A). Direct in vivo expansion of patient-derived xenograft was not possible, as none of the immunodeficient mice developed LMD. In contrast, a cell-derived xenograft that was first expanded in vitro and then injected resulted in LMD. Finally, a comprehensive RNA sequencing analysis of CTCs revealed potential targets, such as the IGF1 signalling pathway.

## 5. Diagnosis

Overall, a definitive diagnosis remains difficult and should include a detailed neurological history and examination, CSF cytology, and magnetic resonance imaging (MRI) of the CNS. In addition, new diagnostic approaches such as the analysis of CTCs and cell-free tumour DNA (ctDNA) are showing promise for earlier detection.

### 5.1. Current Standard in Diagnostics

#### 5.1.1. Symptoms

Most patients with LMD develop symptoms. Therefore, patients with an underlying malignancy who develop symptoms or signs suggestive of multifocal CNS involvement should be investigated immediately. The clinical picture of LMD is often very heterogeneous and depends on the extent and location of the involvement. Typical symptoms include headaches, often due to increased intracranial pressure (ICP), cranial nerve involvement, seizures, somnolence, confusion, and meningism. Spinal symptoms may include radicular pain, dermatomal paraesthesia, bowel or bladder dysfunction, and limb weakness. Fever, photophobia, and meningism are extremely rare in patients with LMD, in contrast to patients with bacterial or haemorrhagic meningitis [28].

#### 5.1.2. CSF

The “gold standard” for diagnosis of LMD is the presence of tumour cells in CSF (Figure 2). In addition, increased opening pressure during lumbar puncture (LP), a low CSF glucose concentration, a high CSF protein concentration, and leucocytosis with lymphocytic pleocytosis, often without identification of malignant cells, are frequently observed [28]. As the sensitivity of an initial LP is estimated to be only 44–67% [29,30,31,32] it may be helpful to repeat LP up to three times to increase sensitivity (80–90%) if LMD is suspected and MRI is not diagnostic [33]. LP should be performed at least two weeks after craniotomy to avoid false-positive cytology [34]. If possible, a large volume sample of 10–20 mL of CSF should be obtained.

#### 5.1.3. MRI

In addition to LP, patients with suspected LMD should also undergo MRI of both the brain and spine. It is generally recommended that LP is performed after imaging to reduce misinterpretation of leptomeningeal enhancement due to iatrogenic manipulation [28].

LMD may present as both nodular and/or curvilinear enhancement in the cortical sulci of the cerebrum and in the folia cerebelli (Figure 3a). LMD can also affect the cranial nerves and basal pons. In the spine, it may present as smooth and nodular enhancement along the pia mater of the spinal cord, involving the nerve roots of the cauda equina (Figure 3b) [35]. In the case of suspicious MRI findings in the absence of clinical symptoms or CSF findings without abnormalities, MRI should be repeated in a few weeks and guided by clinical symptoms. For solid tumours with the highest likelihood of CNS metastases or LMD, tumour-specific guidelines recommend CNS imaging in symptomatic patients [36,37]. MRI is generally recommended for this purpose, as it has the highest diagnostic accuracy for detecting brain metastases [38]. The tumour-specific recommendations are as follows:-In MM, high-risk patients (stage IIC and higher) should undergo imaging every six months for the first three years after diagnosis, according to the German guidelines [38]. This interval should be shortened in the presence of locoregional or distant metastases.-In breast cancer, brain imaging should not be routinely performed in all asymptomatic patients at initial diagnosis of metastases or during disease surveillance [37]. In some subtypes (asymptomatic HER2-positive breast cancer or triple negative breast cancer), brain metastases are more common at the initial diagnosis of metastases. This may justify subtype-specific brain imaging in asymptomatic patients with metastatic breast cancer.-Patients with small cell lung cancer should receive prophylactic cranial irradiation (PCI) if they are in remission after completing chemo-radiotherapy [36]. In patients who have not received PCI, the ESMO guidelines recommend regular brain MRI [39,40]. However, the use of PCI does not appear to have any effect on the development of LMD [41].-After successful curative therapy, imaging is not recommended for the detection of brain metastases in clinically normal patients with non-small cell lung cancer (NSCLC), as there are currently no clinical data on outcomes [36]. However, advanced NSCLC has a very high metastatic potential: In stage III, in addition to the relatively high risk of locoregional recurrence and the risk of developing distant metastases, there is also a high risk of developing brain metastases. In addition to systemic metastases outside the CNS, stage III patients have a cumulative risk of up to 50% of developing brain metastases at five years [42,43].

### 5.2. Novel Perspectives in Diagnostics

Numerous new approaches are under clinical investigation to improve diagnosis and monitor response to treatment. One method is the analysis of so-called “liquid biopsies”, which can be used to collect and measure tumour components such as CTCs, ctDNA, and cell-free RNA (cfRNA), as well as exosomes in body fluids [44].

#### 5.2.1. Circulating Tumour Cells (CTCs)

Several semi-automated cellular assays have been developed to improve the detection of CTCs in CSF for the diagnosis of LMD [45,46,47,48]. The assays include immunoflow analyses using fluorescently labelled antibodies against membrane-bound tumour cell proteins, such as CD146, a human high molecular weight melanoma-associated antigen (HMW-MAA), also known as melanoma chondroitin sulphate proteoglycan (MCSP) in MM [46,47], and others such as the epithelial cell adhesion molecule (EpCAM), which is not expressed on MM cells [49]. Positive tumour cells are therefore detected by positive expression of tumour cell proteins and the nuclear marker DAPI, as well as negative expression of leukocyte markers (CD45). Some approaches appear promising with higher sensitivity at first LP compared to conventional cytology (75–100% vs. 44–46%) [50]. In a large cohort of 95 patients (36 breast, 31 lung, and 28 other), LMD was diagnosed by CSF CTC measurement with a sensitivity of 93% and a specificity of 95% [45]. The study was designed to enrol patients with epithelial tumours who were suspected of having LMD, either because of clinical symptoms or MRI findings. The diagnosis of LMD was made by conventional diagnostic techniques using CSF cytology and/or MRI and LMD was detected in 32% of the patients at the initial evaluation. Using ROC analysis, a cut-off value of ≥1 CSF CTC/mL provided the best threshold for the diagnosis of LMD, achieving the sensitivity and specificity mentioned above.

In addition to advantages at initial diagnosis, the ability to quantitatively count the number of CTCs in CSF can also provide an assessment of disease burden, mortality, and/or therapeutic response. A retrospective study of 101 patients with LMD who underwent CSF CTC quantification with the CellSearch^®^ platform could predict a doubling of mortality risk at the optimal cut-off of ≥61 CSF CTCs/3 mL [48].

Nevertheless, in terms of clinical application, CTC assays for the diagnosis of LMD require cautious interpretation. For MM, only data from a few samples are available and larger studies primarily include patients with breast and lung cancer. Moreover, on the one hand, there can be false-negative results, in part because only 85% of MM cells express HMW-MAA/MCSP [47] and epithelial tumours may lose EpCAM expression as they transition to a mesenchymal subtype [51]. On the other hand, false-positive results are also possible, for example due to brain parenchymal metastases close to the CSF compartment, which may shed CTCs into CSF, as mentioned by Lin et al. [45].

#### 5.2.2. Cell-Free Tumour DNA (ctDNA)

Most cell-free DNA (cfDNA) is derived from normal body cells or food intake [52] and only a small fraction, <1% of cfDNA, is associated with tumours [53], which are short, double-stranded fragments of tumour DNA shed by tumour cells as a result of cell apoptosis and/or necrosis [54]. Plasma-detected ctDNA has shown promise for early diagnosis and tumour characterisation [55].

For brain tumours and LMD, the analysis of plasma ctDNA is suboptimal, as only low amounts of brain-derived ctDNA are detectable due to the BBB [56]. In this regard, several trials have demonstrated the superiority of ctDNA from CSF, which yielded fewer but predominantly tumour-derived cfDNA [44,57,58]. Ying et al. compared the ctDNA of matched CSF and plasma samples from 72 advanced NSCLC patients with confirmed LMD by using a panel of 168 lung cancer-related genes [58]. Mutation detection rates (81.5% vs. 62.5%; *p* = 0.008) and the maximum allelic fraction (43.6% vs. 4.6%; *p* < 0.001) of CSF vs. plasma demonstrated superior mutation identification and genomic analysis of LMD in CSF.

Another trial compared CSF ctDNA by droplet digital PCR (ddPCR) and next-generation sequencing with cytology and MRI in seven MM patients with LMD [59]. There was a strong correlation between positive ddPCR results, tumour cells on cytology, and abnormalities on MRI. In addition, positive ddPCR results were found in approximately 30% of CSF samples that had either no or questionable tumour cells on cytology. As the correlation between positive ctDNA and abnormal MRI was stronger than that between positive cytology and MRI, the authors concluded that ctDNA analysis may be the superior approach for diagnosing LMD in MM patients.

In addition to early detection, ctDNA analysis may also be used to assess treatment response and monitor tumour burden. Variant allele frequency (VAF) is the number of mutant molecules relative to the total number of wild-type molecules at a given location in the genome. Several data suggest that the response to systemic therapy in patients with advanced disease is positively associated with changes in VAF and ctDNA. Janku et al. measured B-Raf protooncogene, serine/threonine kinase (*BRAF*) mutations in ctDNA from formalin-fixed paraffin-embedded tumour or plasma samples [60], derived from advanced cancer or malignant histiocytosis with known *BRAF^V600^* status. It was shown that a decrease in the percentage of *BRAF^V600^* ctDNA, compared to an increase or no change, was associated with a longer time to treatment failure (10.3 vs. 7.4 months; *p* = 0.045). Similarly, a phase 2 trial of tebentafusp in patients with metastatic uveal melanoma (NCT02570308) showed a significant association between reduction in ctDNA and OS (*p* = 8.89 × 10^−7^) [61]. Wijetunga et al. calculated the VAF in CSF ctDNA in 14 patients with LMD from solid tumours before and after undergoing proton craniospinal irradiation [62]. Higher mean VAF before and after irradiation were both significantly associated with worse OS (*p* = 0.05 and *p* = 0.008, respectively).

However, as parenchymal lesions adjacent to the leptomeningeal space or the ventricular system may also release ctDNA without vital malignant cells into CSF, interpretation can therefore be difficult [35]. White et al. demonstrated improved sensitivity and accuracy in the diagnosis of LMD with ctDNA in CSF compared to cytological analysis (94% vs. 75%; *p* = 0.02) in patients with malignancies not adjacent to CSF [63]. Nevertheless, in three patients with parenchymal brain metastases neighbouring CSF and no evidence of LMD, ctDNA analysis was positive in all patients, whereas cytology was negative in all patients. Therefore, interpretation of the results requires caution due to the possibility of false-positive results as CSF-neighbouring tumours may contaminate the sample.

#### 5.2.3. Cell-Free RNA (cfRNA)

Multiple challenges remain with using either CTCs or ctDNA for the diagnosis or monitoring of LMD. Isolating these cells is technically challenging and requires large amounts of material [64]. Furthermore, in MM, ctDNA analysis can only be used for patients with MM-specific mutations, such as *BRAF^V600^*, *NRAS^Q61^*, or *TERT^prom^*, and the combination of hotspot gene alterations only covers approximately 80% of MM cases [65]. In this context, Albrecht et al. identified cfRNA biomarkers (*KPNA2*, *DTL*, *BACE2*, and *DTYMK*) that were significantly higher in MM patient plasma compared to healthy donor plasma (*p* < 0.0001) [66]. In addition, there were no significant differences in cfRNA copy numbers between different mutational subgroups. Li et al. performed a comprehensive RNA analysis in CSF samples from NSCLC LMD patients using single LMD cell detection, RNA sequencing, transcriptome analysis, and multiplexed microfluidic cfRNA real-time quantified PCR analyses [67]. They were able to detect tumour-associated cfRNA (e.g., *CEACAM6*) in the CSF of NSCLC patients that matched the gene expression profile in LMD cells and were distinctly different from the cfRNA detectable in healthy controls. To date, the utility of cfRNA for the diagnosis of LMD in MM has not been reported.

## 6. Therapy

The development of new therapeutic strategies has revolutionised the treatment of advanced MM, including patients with brain metastases (MBMs) [68], culminating to date in a three-year OS of 71.9% for combined immune checkpoint inhibitors (ICIs) with ipilimumab plus nivolumab in patients with asymptomatic MBM [69]. In contrast, survival in patients with LMD has not changed drastically over recent decades and is still typically measured in weeks to a few months [28]. Published survival data range from 1.7 to 2.5 months for MM, 1.75 to 4.5 months for breast cancer, and 3 to 6 months for lung cancer, with one-year survival rates of 7% (MM), 16–24% (breast cancer), and 19% (lung cancer), respectively [70,71,72,73,74,75,76,77,78,79,80,81,82,83,84,85,86,87].

Possible reasons for this still poor survival could include:Diagnosis remains challenging, as outlined above.At the time of LMD diagnosis, most patients have been exposed to various drugs, specifically ICIs and targeted therapies. LMD cells might represent a subpopulation of resistant cells in a “sheltered” TME.In contrast to parenchymal metastasis, local tumour control with stereotactic radiotherapy (RTx) is often not possible due to the distribution of LMD.Studies suggest a reprogramming of the LMD TME with a dysfunctional T cell landscape, making systemic therapy less effective [25].While we are seeing an increase in clinical trials for patients with brain metastases from various tumour types, dedicated clinical trials for LMD patients are largely absent.LMD often leads to rapid decline and significant morbidity, often resulting in the recommendation of supportive care only.

While treatment options for LMD remain limited, contemporary treatment modalities can be divided into systemic, IT, and radiation-based approaches.

### 6.1. Systemic Therapy

#### 6.1.1. Chemotherapy

Treatment of brain tumours with chemotherapy is limited, mainly because the BBB poses challenges for drug penetration with most systemic therapies [88]. The alkylating cytostatic temozolomide has been evaluated as an option for systemic chemotherapy. However, in a phase 2 trial, the median survival in 19 patients with solid tumours (breast cancer and NSCLC) and LMD was only 43 days (95% CI, 28.7–57.3) [89].

#### 6.1.2. Immunotherapy

Due to their molecular structure, the general consensus was that ICIs are not able to cross the BBB. Using murine models, it has been suggested that systemic therapy with ICIs leads to an intracranial response only in the presence of extracranial metastases, which promote the activation and release of CD8+ cells in the periphery [90]. By analysing intra-tumoural CD8^+^ T cells from parenchymal brain metastases, ICIs dramatically increased the migration of CD8+ T cells into the brain (14-fold), rather than enhancing the activation and dissemination of CD8+ T cells. The systematic review by van Bussel et al. analysed the pharmacodynamics and pharmacokinetics of nivolumab and ipilimumab and proclaimed the possibility of ICI crossing the BBB by two mechanisms [91]. ICIs can either bind to PD-1 or CTLA-4 on peripheral lymphocytes, which subsequently enter the CNS (mechanism 1), or penetrate the BBB directly (mechanism 2). Both nivolumab and ipilimumab are IgG monoclonal antibodies with the neonatal fragment crystallisable receptor (FcRn). IgG antibodies with FcRn can enter cells such as macrophages in the choroid plexus and enter CSF by endocytosis via FcRn-mediated transcytosis [92]. In vivo, the CSF levels of nivolumab were measured and quantified by enzyme-linked immunosorbent assays in a cohort of MM patients with suspicion of LMD [93]. The nivolumab concentrations ranged from 35 to 150 ng/mL with a CSF to serum ratio of 52–299, indicating low penetration of nivolumab into the brain. In another trial, measurement of the serum-to-CSF ratio of trastuzumab in patients with HER2-positive breast cancer (start 420:1) showed an increase in trastuzumab after RTx (76:1) or LMD (49:1), indicating a disrupted blood–CSF barrier and therefore a potential for agents to enter this space [94]. Prakadan et al. used single-cell RNA and cfDNA sequencing in longitudinal CSF samples from LMD patients undergoing ICI treatment to investigate molecular characteristics [95]. They were able to show that CD8+ cells in the CSF of ICI samples were more numerous and proliferative with increased expression of proliferation genes such as *MKI67*, *BIRC5*, and *TOP2A* compared to the baseline CSF. In addition, higher levels of IFN-γ signalling suggest a possible modulation of the immune landscape in CSF during ICI therapy.

For combined ICI therapy with ipilimumab and nivolumab, within the CheckMate 204 trial, OS in patients with asymptomatic MBM was 71.9% at three years [69]. However, patients with LMD were not included in this study. Similarly encouraging to the CheckMate 204 study, the ABC trial showed an OS of 51% after five years of follow-up in asymptomatic MBM patients with the same combination (cohort A). [96]. Moreover, this trial also included MM patients with LMD (cohort 4). None of the four LMD patients enrolled showed a response to nivolumab monotherapy. In a large retrospective analysis of 178 LMD patients, ICI monotherapy in MM patients with LMD (12/178) also showed no survival benefit (HR 1.2; *p* = 0.59); however, the majority of patients had prior exposure to ICIs [2].

Three recent studies described the outcomes of patients with LMD from various malignancies, treated with ICIs, either with the combination of ipilimumab and nivolumab [97] or the single agent pembrolizumab [98,99]. Importantly, only two melanoma patients were included across these three trials. The study with combination treatment of ipilimumab and nivolumab enrolled 18 patients with various malignancies and LMD and met its primary endpoint with 44% of patients (8/18) alive at three months. The median follow-up based on patients still alive was 8.0 months (range, 0.5–15.9 months). Monotherapy with pembrolizumab achieved a CNS response, defined as either radiological, cytological, or clinical, in 38% of patients at week 12 [99]. The study cohort consisted of 16 heavily pre-treated patients with various solid tumours (hormone receptor-positive breast cancer, high-grade glioma, NSCLC, head and neck cancer, and cutaneous squamous cell carcinoma) and LMD. Although not statistically significant, the OS of patients with positive cytology was worse than that of patients with no detectable tumour cells on cytology (3.7 months vs. 10.3 months, log-rank *p* = 0.29). The median OS was 4.9 months. The second single-agent study with pembrolizumab, which included 20 patients with also various solid tumours (breast, lung, and ovarian), met its primary endpoint, with 12 of 20 patients alive at three months after enrolment [98]. The investigator designed a Simon two-stage approach, comparing a null hypothesis of 18% of patients alive at three months with an alternative of 43%.

#### 6.1.3. Targeted Therapy

The discovery of mutations in the *BRAF* gene in human cancer cells in 2002 laid the foundation for targeted therapy in MM [100]. There are several case reports describing the outcome of MM patients with LMD treated with targeted therapy. Some of the results are very encouraging, as OS rates of 11 months and longer have been reported [101,102,103], but most of these patients were treatment-naïve to BRAF inhibitor monotherapy and/or dual therapy with BRAF inhibitors and mitogen-activated protein kinase kinase (MEK) inhibitors at the time of LMD diagnosis. However, once LMD develops while on or after prior exposure to targeted therapy, survival remains poor [2]. Numerous studies have investigated the mechanisms of escape from BRAF inhibition [104]. Several lines of evidence suggest that BRAF inhibitors may have limited penetration into the brain and leptomeningeal lesions due to active drug efflux transporters [105]. Of particular note are two efflux transporters, P-glycoprotein (P-gp, MDR1, ABCB1) and breast cancer resistance protein (BCRP, ABCG2) [106,107], members of the ATP-Binding Cassette (ABC) and Solute Carrier (SLC) families [108]. Preclinical data strongly support an interaction between the targeted therapy and P-gp and BCRP [105,109,110,111], resulting in an intracranial concentration of less than 10% of the plasma concentration [112]. In a study of six patients, significant variations in CSF concentrations of the BRAF inhibitor vemurafenib were observed, which may be due to differences in BBB integrity following previous local treatments such as surgery or RTx [113]. In conclusion, the development of targeted therapy with reliable CNS penetration requires further investigation.

Prospective studies have already been initiated, such as a monocentric study in patients with MBM and/or LMD at the M.D. Anderson Cancer Center to receive a high-dose regimen of encorafenib and binimetinib to achieve higher intracranial drug concentrations (NCT05026983) [114]. A phase 1 clinical trial is currently evaluating a novel CNS-penetrant BRAF inhibitor, PF-07284890, in combination with binimetinib, for the treatment of *BRAF^V600^* mutant solid tumours with or without CNS involvement or LMD (NCT04543188) [115]. In addition, preclinical data on a new BRAF inhibitor (compound Ia) suggest that it may provide exceptional intracranial results due to limited P-gp-mediated efflux and its lower molecular weight, which may allow it to penetrate the brain more efficiently [116].

### 6.2. Intrathecal Therapy

#### 6.2.1. Chemotherapy

Most chemotherapeutic agents are hydrophilic or of high molecular weight, with correspondingly poor CNS penetration, so IT administration is preferred. Overall, efficacy in MM with LMD is limited. The largest case study included only nine patients who received IT chemotherapy with thiotepa and topotecan, with a median OS of only eight weeks; however, two patients had a median OS of 104 weeks [117].

#### 6.2.2. Interleukin-2

With low levels of T cells present in CSF, IT interleukin-2 (IL-2) administration has been used in order to generate an immune response against immunogenic tumour cells in CSF. A retrospective review analysed a cohort of 43 LMD patients treated with IT IL-2; median OS from initiation of therapy was 7.8 months, with one-, two-, and five-year OS rates of 36%, 26%, and 13%, respectively [118]. All patients developed adverse events (AEs), including fever, chills, and symptoms of increased ICP, such as nausea and headaches. Other symptoms were vomiting and transient changes in mental status, which resolved when IT IL-2 dosing was delayed and/or reduced. All patients required additional CSF drainage to control symptoms and reduce ICP during the induction period. The authors concluded that despite promising responses in a subset of MM patients with LMD, this setting should be reserved for patients with excellent performance status at treatment initiation and should only be performed in specialised medical centres able to manage toxicity.

#### 6.2.3. Immunotherapy

As mentioned above, immunotherapy with IL-2 has shown encouraging results, but both patients and physicians often have to deal with serious side effects. The group led by Glitza et al. has therefore sought to harness the benefits of immunotherapy and adapt it to reduce toxicity [119]. Based on preclinical results, a phase 1/1b study (NCT03025256) was designed for 25 MM patients with LMD receiving concurrent IT and intravenous (IV) nivolumab. Notably, the cohort was predominantly a poor prognosis group, with 92% and 88% of participants enrolled having received prior treatment (median, 2; range, 1–6) and progressing on prior ICI therapy (anti-PD-1 with or without anti-CTLA-4), respectively. Initially, only IT administration without IV administration was used to identify method-specific AEs, which included nausea, dizziness, and vomiting (all grade 1) and neck pain in one patient (grade 2). As there were no dose-limiting toxicity at any IT level during the escalation phase (5, 10, and 20 mg), the protocol was amended to provide 50 mg of nivolumab, with continuation of mainly grade 1 or 2 AEs. At the recommended dose of 50 mg of IT and 240 mg of IV nivolumab every two weeks, a median OS of 4.9 months was achieved, with OS rates of 44% and 26% at 26 and 52 weeks, respectively. The authors concluded that the IT approach with nivolumab in patients with LMD is both safer than IT IL-2 and shows encouraging results even in heavily pre-treated patients.

### 6.3. Radiotherapy (RTx)

RTx is considered palliative and can be used in patients with LMD to address neurological symptom burden [120].

The standard modalities used are whole-brain radiation therapy (WBRT) and local RTx to relieve symptoms in patients with focal involvement, most commonly utilised in patients with spinal cord involvement. Recent advances with the goal of reducing neurocognitive decline techniques such as hippocampus-sparing irradiation and concurrent treatment with cholinesterase inhibitors such as memantine have shown encouraging results in prospective studies [121,122]; however, this approach is often not applicable in LMD patients because of the often diffuse meningeal involvement.

Proton craniospinal irradiation has already shown impressive results in a phase 2 trial, with improved OS and progression-free survival in patients with LMD compared to the standard regimen of photon involved-field RTx [123]. Protons also have the advantage of emitting most of their energy in the last few millimetres of their range, which means that, unlike photon radiation, the front organs are not exposed to radiation [124]. A monocentric, single-arm trial is currently evaluating proton craniospinal irradiation in patients with LMD from both solid and haematological cancers (NCT05746754) [125].

### 6.4. Novel Perspectives in Treatment

A number of trials are underway to modify or test existing systemic therapies, develop new drugs, and adapt local tumour control with RTx to improve the treatment of LMD (Table 1).

Modifications of already existing systemic therapies include the use of immunotherapy in a multicentre, single-arm setting to evaluate the IT combination of a fixed dose of nivolumab and an escalating dose of ipilimumab with concurrent treatment with systemic nivolumab and ipilimumab (NCT05598853) [126]. Another approach is to combine pembrolizumab with the tyrosine kinase inhibitor lenvatinib, which is already approved in several malignancies (thyroid cancer, hepatocellular carcinoma, and endometrial cancer) and works by blocking pro-angiogenic receptors (NCT04729348) [127]. In MM, although the first-line combination of ICI plus lenvatinib failed to improve OS and had to be discontinued (LEAP-003 trial; NCT03820986) [128], a second-line approach in previously treated patients (LEAP-004; NCT03776136) [129] resulted in durable responses [130]. A diagnosis of LMD was an exclusion criterion in both trials. In addition to immunotherapy, a modified dosage of targeted therapy is used at the M.D. Anderson Cancer Center with high-dose encorafenib and binimetinib in patients with MBM and/or LMD (NCT05026983) [114]. Another trial is randomly comparing triple therapy with encorafenib, binimetinib, and nivolumab with ipilimumab and nivolumab in patients with MBM. Participants may have LMD (NCT04511013) [131]. The MEK1 and mitogen-activated protein kinase/extracellular signal-regulated kinase kinase kinase-1 (MEKK1) inhibitor E6201 showed brain distribution characteristics that were minimally affected by P-gp and BCRP efflux transport and demonstrated an exceptional response in an MM patient with intra- and extracranial metastases [132,133]. This compound is currently being tested in combination with the BRAF inhibitor dabrafenib in a phase 1 trial in patients with MBM (NCT05388877) [134]. Due to the frequent diffuse seeding of tumour cells in LMD, local tumour control with RTx is often inadequate and predominantly occurs in a palliative setting. Several studies are investigating different modes of action either to increase the likelihood of radiation hitting tumour cells or in combination with systemic therapy. IT brachytherapy using rhenium-186 nanoliposomes has exhibited promising results in preclinical studies and is currently undergoing evaluation in phase 1 trials for patients with LMD (ReSPECT-LM, NCT05034497). This approach aims to deliver localised RTx directly to the leptomeninges and has already been tested in the ReSPECT-GBM trial (NCT01906385) [135] in patients with recurrent glioma with no observed dose-limiting toxicities. IT administration in non-tumour-bearing rats has been shown to be free of significant toxicity and result in prolonged survival in two LMD models [136]. As already mentioned above, proton craniospinal irradiation has shown promising results in terms of OS and reduced radiation exposure to healthy tissue [123]. At the moment, a monocentric trial is open for patients with LMD from both solid and haematological cancers (NCT05746754) [125]. The established setting of WBRT is combined with the novel agent silibinin, a natural polyphenolic flavonoid isolated from seed extracts of the herb milk thistle (Silybum marianum), in patients with solid tumours and multiple brain metastases and/or LMD (NCT05793489) [137]. Recent studies have shown that silibinin is able to impair STAT3 activation [138], which induces and maintains a pro-metastatic landscape by a subpopulation of reactive astrocytes surrounding metastases [139]. Another new agent is the antimetabolite pemetrexed as IT single agent versus IT pemetrexed with concurrent involved field RTx (NCT05305885) [140]. Pemetrexed, which targets three enzymes—thymidylate synthase, dihydrofolate reductase, and glycinamide ribonucleotide formyl transferase—has shown promising results in patients with advanced NSCLC and LMD who have failed multiple lines of treatment [141]. Lastly, it has been shown that tumour cells in CSF express the iron-binding protein lipocalin-2 (LCN2) and its receptor SCL22A17 to outcompete other cells, such as macrophages, for free iron and thus ensure tumour metabolism [142]. These results have been confirmed in a murine model where IT injections of an iron chelator (deferoxamine) reduced LMD growth and showed a significant increase in survival. A phase 1a/b trial is underway in the US evaluating IT injections of deferoxamine in patients with solid malignancies and LMD (NCT05184816) [143].

## 7. Conclusions

LMD continues to pose many challenges for diagnosis and treatment and the prognosis for these patients remains poor, despite contemporary treatment options. In the absence of highly sensitive diagnostic tools, patients with underlying malignancy and multifocal neurological symptoms should be promptly evaluated. In addition, regular CNS imaging (preferably with MRI) is desirable from the time of distant metastasis in high-risk tumours.

Further studies are needed to better understand the pathophysiology of the disease and to develop new, urgently needed therapies. Overall, more trials specifically designed for LMD are needed, or LMD patients should be included even in early phases of clinical trials for advanced MM.

## Figures and Tables

**Figure 2 ijms-24-11443-f002:**
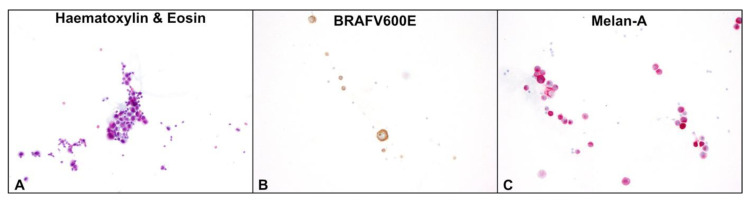
Cerebrospinal fluid with tumour cells (magnification, 200×). (**A**) Haematoxylin and eosin stain with large, atypical tumour cells and small round lymphocytes. (**B**) Tumour cells with positive BRAF^V600E^ marker (brown) and unstained lymphocytes in immunohistochemistry (IHC). (**C**) Tumour cells with positive Melan-A marker (red) and unstained lymphocytes in IHC.

**Figure 3 ijms-24-11443-f003:**
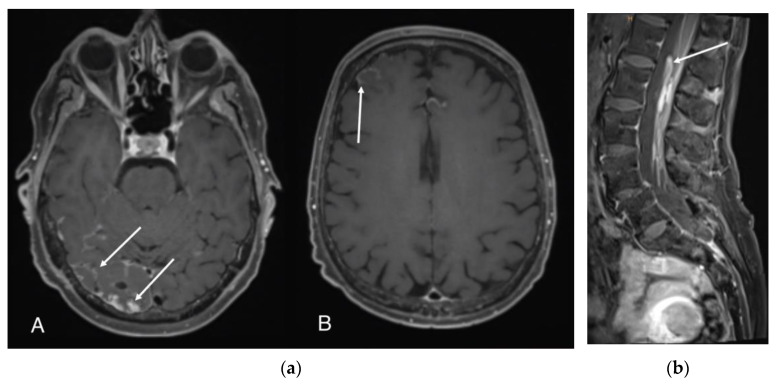
(**a**). Postcontrast T1-weighted images demonstrate nodular enhancement (A) and curvilinear leptomeningeal enhancement (B). (**b**). Gadolinium-enhanced sagittal T1 sequence reveals nodular leptomeningeal spinal metastases and shows enhancement and clumping of the cauda equina nerve roots.

**Table 1 ijms-24-11443-t001:** Novel perspectives and current clinical trials for patients with LMD. CTI, ClinicalTrials.gov identifier; IT, intrathecal; LMD, leptomeningeal disease; MC, multi-centre; NA, not available; NR, non-randomised; NSCLC, non-small cell lung cancer; P, participants; R, randomised; SC, single-centre; SYS, systemic; WBRT, whole-brain radiation therapy.

Title	CTI	Phase	P	Design	Disease	Intervention	Country
Intrathecal Double Checkpoint Inhibition (IT-IO)	NCT05598853	1	26	NR, MC	NSCLC and melanoma	IT/SYS nivolumab + ipilimumab	Switzerland
Intrathecal Application of PD1 Antibody in Metastatic Solid Tumors With Leptomeningeal Disease (IT-PD1/NOA 26) (IT-PD1)	NCT05112549	1	46	NR, MC	Solid tumours	Part 1: Dose escalation of IT nivolumab in 4 cohorts Part 2: Dose expansion of IT nivolumab	Germany
Pembrolizumab And Lenvatinib In Leptomeningeal Metastases	NCT04729348	2	19	NR, MC	Solid tumours	Pembrolizumab + lenvatinib	USA
Binimetinib and Encorafenib for the Treatment of Metastatic Melanoma and Central Nervous System Metastases	NCT05026983	2	35	NR, SC	Melanoma	Binimetinib + encorafenib high dose	USA
A Study to Compare the Administration of Encorafenib + Binimetinib + Nivolumab Versus Ipilimumab + Nivolumab in BRAF-V600 Mutant Melanoma with Brain Metastases	NCT04511013	2	112	R, MC	Melanoma	Arm A: Encorafenib, binimetinib + nivolumab Arm B: Ipilimumab + nivolumab	USA
E6201 and Dabrafenib for the Treatment of Central Nervous System Metastases from BRAF V600 Mutated Metastatic Melanoma	NCT05388877	1	18	NR, MC	Melanoma	MEK-1/MEKK-1 inhibitor E6201 + dabrafenib	USA
Proton Cranio-spinal Irradiation for Leptomeningeal Metastasis (CSI ProLong)	NCT05746754	2	50	NR, SC	Solid tumours and haematological cancer	Proton radiotherapy with 30 Gy in 10 fractions to the entire craniospinal axis	Denmark
Intraventricular Administration of Rhenium-186 NanoLiposome for Leptomeningeal Metastases (ReSPECT-LM)	NCT05034497	1	18	NR, MC	LMD of any primary type	Single-dose Rhenium-186 NanoLiposome (186RNL)	USA
Prospective Double Arm Randomized Trial: WBRT Alone and WBRT Plus Silibinin	NCT05793489	NA	44	R, SC	Solid tumours with brain metastases and/or LMD	Arm A: WBRT + silibinin Arm B: WBRT	Italy
Intra-pemetrexed Alone or Combined With Concurrent Radiotherapy for Leptomeningeal Metastasis	NCT05305885	NA	100	R, MC	Solid tumours	Arm A: IT pemetrexed in combination with involved field RTX Arm B: IT pemetrexed monotherapy	China
A Study of Deferoxamine (DFO) in People With Leptomeningeal Metastasis	NCT05184816	1a/1b	35	NR, SC	1a: Solid tumours 1b: NSCLC	Phase 1a: Dose escalation of IT deferoxamine (solid tumours) Phase 1b: Dose expansion (NSCLC)	USA

## Data Availability

No new data were created or analyzed in this study. Data sharing is not applicable to this article.

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
