# Peer review of "Leptomeningeal Metastases in Melanoma Patients: An Update on and Future Perspectives for Diagnosis and Treatment"

_ijms, 2023, doi:10.3390/ijms241411443_

Round 1

Reviewer 1 Report

Dear colleagues,

Thank you very much for giving me the opportunity to go through your manuscript I read with great interest. 

I found the manuscript very interesting, easy to read and clinically highly relevant.

However, I believe there are few issues you could adress to improve the quality of your manuscript and discussion.

1. I would suggest to remove the material and methods sections from the current place (just before the discussion) and move it back to where it usually belongs (after introduction).

2. The material and methods section is currently really poor. Please provide the used key words, time frame (publications from the last 3 or 10 years?), the database used, etc.

3. In the diagnostic section. Could you be more specific with regards to imaging? I was missing the reference to international guidelines of the most relevant tumor types through the manuscript and the references list particularly when it comes to diagnostic. As a matter of fact, you correctly reported the tumors most frequenlty associated with LMM (leptomeningeal metastases). In case of melanoma for instance, the current guidelines recommend cMRI from a certain stage. Could you provide further informations based on international guidelines with regards to the indication for brain imaging in the most relevant tumor types leading to LMM. A certain algorithm may help a provide a better/earlier detection of LMM. 

4. In the therapy section "survival has not changed" is not specific and needs to be improved. Please if provide number to quantify this statement for the reader with regards to the different types of cancers leading to LMM.

I am looking forward to the revised version of your manuscript.

Best regards.

Reviewer 2 Report

Leptomeningeal disease (LMD) is a devastating complication of cancer with a particularly poor prognosis. Malignant melanoma (MM) has one of the highest rates of metastasis to the leptomeninges, with approximately 10-15% of patients with advanced disease developing LMD. Despite advances in treatment options, the incidence of LMD appears to be increasing and current treatment modalities have a limited impact on survival. While a 71.9% 3-year overall survival (OS) was achieved with ipilimumab plus nivolumab in patients with asymptomatic melanoma brain metastases (MBM), in MM patients with LMD the OS was only 3.5 months.

This is a well-documented and well-written paper with 118 references, three educating figures, and one table on novel perspectives and current clinical trials for patients with LMD. The authors review the anatomical structure of the brain, the tumor microenvironment, the biology of LMD, the diagnosis and therapy, including chemotherapy, immunotherapy, targeted therapy, intrathecal therapy, novel treatment perspectives, and the ongoing clinical trials of LMD in MM patients.

With such a wealth of important information, the paper will help the oncologic community to appreciate the diagnostic and therapeutic challenges of this disease whose prognosis remains poor despite contemporary treatment options.

Author Response

Dear Reviewer,

Thank you very much for your positive feedback on our manuscript. We appreciate your recognition of the importance of this disease and its challenges.

We are grateful that you found our work well documented and helpful to the oncology community in understanding the diagnostic and therapeutic complexities of leptomeningeal disease in malignant melanoma. Your feedback encourages us to continue addressing this critical issue.

Thank you for your time and valuable input. We look forward to receiving your feedback on the revised version of our manuscript. To help you identify the changes, we have highlighted them throughout the document.

Best regards,
Julian Steininger
Friedegund Meier
Isabella C. Glitza Oliva